# Depo Medroxyprogesterone (DMPA) Promotes Papillomavirus Infections but Does Not Accelerate Disease Progression in the Anogenital Tract of a Mouse Model

**DOI:** 10.3390/v14050980

**Published:** 2022-05-06

**Authors:** Jiafen Hu, Sarah A. Brendle, Jingwei J. Li, Vonn Walter, Nancy M. Cladel, Timothy Cooper, Debra A. Shearer, Karla K. Balogh, Neil D. Christensen

**Affiliations:** 1The Jake Gittlen Laboratories for Cancer Research, College of Medicine, Pennsylvania State University, Hershey, PA 17033, USA; sab40@psu.edu (S.A.B.); jenny.jingwei.li@gmail.com (J.J.L.); ncladel@gmail.com (N.M.C.); dshearer@pennstatehealth.psu.edu (D.A.S.); kkb4@psu.edu (K.K.B.); ndc1@psu.edu (N.D.C.); 2Department of Pathology, College of Medicine, Pennsylvania State University, Hershey, PA 17033, USA; 3Department of Public Health Sciences, College of Medicine, Pennsylvania State University, Hershey, PA 17033, USA; vwalter1@pennstatehealth.psu.edu; 4Department of Biochemistry and Molecular Biology, College of Medicine, Pennsylvania State University, Hershey, PA 17033, USA; 5Integrated Research Facility at Fort Detrick, National Institute of Allergy and Infectious Diseases, NIH, Fort Detrick, Frederick, MD 21702, USA; timothy.cooper@nih.gov; 6Department of Microbiology and Immunology, College of Medicine, Pennsylvania State University, Hershey, PA 17033, USA

**Keywords:** contraceptive, progesterone, Depo-Provera, DMPA, 17β-estradiol, lower genital tract, mouse papillomavirus model, viral persistence, anti-viral, cytokines, tumor progression

## Abstract

Contraceptives such as Depo-medroxyprogesterone (DMPA) are used by an estimated 34 million women worldwide. DMPA has been associated with increased risk of several viral infections including Herpes simplex virus-2 (HSV-2) and Human immunodeficiency virus (HIV). In the current study, we used the mouse papillomavirus (MmuPV1) anogenital infection model to test two hypotheses: (1) contraceptives such as DMPA increase the susceptibility of the anogenital tract to viral infection and (2) long-term contraceptive administration induces more advanced disease at the anogenital tract. DMPA treatments of both athymic nude mice and heterozygous NU/J (Foxn1^nu/+^) but ovariectomized mice led to a significantly increased viral load at the anogenital tract, suggesting that endogenous sex hormones were involved in increased viral susceptibility by DMPA treatment. Consistent with previous reports, DMPA treatment suppressed host anti-viral activities at the lower genital tract. To test the impact of long-term contraceptive treatment on the MmuPV1-infected lower genital tract, we included two other treatments in addition to DMPA: 17β-estradiol and a non-hormone based contraceptive Cilostazol (CLZ, Pletal). Viral infections were monitored monthly up to nine months post infection by qPCR. The infected vaginal and anal tissues were harvested and further examined by histological, virological, and immunological analyses. Surprisingly, we did not detect a significantly higher grade of histology in animals in the long-term DMPA and 17β-estradiol treated groups when compared to the control groups in the athymic mice we tested. Therefore, although DMPA promotes initial papillomavirus infections in the lower genital tract, the chronic administration of DMPA does not promote cancer development in the infected tissues in our mouse model.

## 1. Introduction

Human papillomavirus (HPV) infection is the most common sexually transmitted disease and claims more than 270,000 women’s lives worldwide from cervical cancer every year [1,2]. The three current prophylactic vaccines offer no protection against pre-existing infections and therefore are unlikely to stop HPV-associated diseases and cancers in HPV positive patients [3]. In addition, low uptake worldwide of these vaccines will leave HPV-associated diseases and cancers as a public health problem until an effective therapeutic treatment is available [4].

Although both men and women are exposed to HPV infections, females have more severe pathology at the anogenital sites [5,6]. This outcome suggests that women may be more prone to viral infection and/or persistence at the anogenital sites. The female menstrual cycle has been implicated in this sex bias [7]. However, whether and how sex hormones play a role in HPV infection and persistence remain unclear. Taking contraceptives is a common family planning method for most women [8]. For example, Depo-medroxyprogesterone (DMPA), a progestin-based contraceptive, is used by an estimated 34 million women worldwide. Whether DMPA plays a role in HPV associated infections and cancers remains elusive to date [9,10]. In addition, HPV associated anal cancer remains a health problem as limited treatment is available especially for immunocompromised HIV/AIDS patients [5,11,12,13]. Interestingly, more anal cancer incidences were reported in women than in men [1,14]. We hypothesized that contraceptive usage by women might increase HPV infections at both the lower genital and anal tracts.

Studying the mechanisms of viral pathogenesis of HPVs has been made possible by using several naturally occurring preclinical models (including cow, horse, dog, and rabbit) [15,16]. The mouse papillomavirus (MmuPV1) model provides a promising opportunity to study anogenital infection, as advanced diseases such as cancers have been reported in infected mice at the lower genital tract [17,18,19,20,21,22,23]. Our previous study also demonstrated that viral titers fluctuated during the mouse estrus cycle suggesting that changes in the hormone level may play a role in viral persistence [20]. DMPA has been reported to induce slight thinning of the glycogen vaginal epithelial layer after six months of treatment [24]. DMPA was also found to alter serum cytokine levels including tumor necrosis factor alpha (TNF-alpha), granulocyte colony-stimulating factor (G-CSF), and interleukin 10 (IL-10) in mice [25]. The estrogen receptor has been reported to be associated with cervical cancer in a transgenic mouse model [26]. DMPA has also been associated with higher bacterial and viral load in the lower genital tract [27,28,29,30]. A more recent study demonstrated that UV irradiation together with 17β-estradiol promote carcinogenesis in MmuPV1 infected mice [21]. Previous studies demonstrated that DMPA increased susceptibility of HPV pseudo-virus delivery into the mouse genital tract [31]. However, how DMPA impacts papillomavirus persistence at the anogenital tract and whether long term treatment accelerates cancer development of anogenital tissues is understudied.

The current study tests our two hypotheses: (1) DMPA increases viral susceptibility and persistence in the female anogenital tract by suppressing the local host anti-viral activity; and (2) long-term treatment of contraceptives including DMPA accelerates viral persistence and cancer development at the anogenital tracts. We used Hsd:Nu athymic nude and NU/J heterozygous (Foxn1^nu/+^) mice that have been tested for MmuPV1 infections in our previous studies [18,20,32]. Viral activity was monitored by collecting vaginal lavages and analyzed by qPCR. The infected tissues were harvested for histological, virological, and immunological analyses.

## 2. Materials and Methods

### 2.1. Mice and MmuPV1 Infection

All mouse work was approved by the Institutional Animal Care and Use Committee of Pennsylvania State University’s College of Medicine (COM) and all methods were performed in accordance with guidelines and regulations. The mice were housed (2–3 mice/cage) in sterile cages within sterile filter hoods in the COM BL2 animal core facility. Infectious virus was isolated from lesions on the tails of mice from our previous study [20]. In brief, lesions scraped from the tails of the mice were homogenized in phosphate-buffered saline (1 × PBS) using a Polytron homogenizer (Brinkmann PT10-35) at the highest speed for three minutes while chilling in an ice bath. The homogenate was spun at 10,000 rpm and the supernatant was decanted into Eppendorf tubes for storage at −20 °C. For these experiments, the mouse virus was diluted 1:5 in 1 × PBS and 200 µL and was passed through a 0.2 µm cellulose acetate sterile syringe filter. Viral DNA was quantitated by extraction of the DNA from 5 µL of this stock using qPCR as described later. For each site, 1 × 10^9^ viral DNA equivalents were used for infection.

To test the impact of contraceptives on viral infection, we treated female Hsd: NU outbred nude mice (Foxn1^nu/nu^) (immunocompromised) or NU/J heterozygous (Foxn1^nu/+^) (immunocompetent) with 3 mg of Depo-Provera (Pfizer, Manhattan, New York City, NY, USA) or 5U of pregnant mare’s serum gonadotropin (PMSG, HOR-272, Prospec Bio, East Brunswick, NJ, USA) three days before viral infection without pre-wounding as previously described [20]. To test the impact of long-term treatment of different contraceptives on disease progression in mice, four groups of Hsd:NU outbred nude mice (Foxn1^nu/nu^) (*n* = 6/group) were infected with MmuPV1 as described previously (2). In brief, the lower genital and anal tracts were pre-wounded with doctors’ brush picks coated with Conceptrol (N9, Amazon, Seattle, WA, USA) the day before infection. Twenty-four hours after pre-wounding, the mice were again anesthetized with ketamine and xylazine (100 mg/10 mg/kg) and challenged under anesthesia with infectious MmuPV1 virus (1 × 10^9^ viral DNA equivalents) in the lower genital and anal tracts, respectively. Four weeks post infection, the mice were administered with PBS (group 1), depo-medroxyprogesterone (Depo, DMPA) (group 2), non-hormone based contraceptive Cilostazol (CLZ, Pletal, Otsuka Pharmaceutical Co, Rockville, MD, USA), a safe PDE3A inhibitor, that can block pregnancy without altering the estrous cycle [33,34] (group 3), and 17β-estradiol (0.05 mg/60 days, SE-121, Innovative Research of America, Sarasota, FL, USA, group 4) for up to nine months post infection. DMPA treated mice received 3 mg injection every 30 days. A new 17β-estradiol tablet was inserted every 2 months. Monthly lavages from the lower genital tract and the anal tract were monitored for viral DNA by qPCR.

### 2.2. Vaginal and Anal Lavage for DNA Extraction and qPCR Analysis

Vaginal and anal lavage/swabs were conducted using 25 μL of sterile 0.9% NaCl introduced into the vaginal and anal canals with a disposable filter tip/swab monthly. The qPCR reactions were run in an Agilent qPCR machine as described previously [20]. All samples were tested in duplicates or triplicates [32]. Viral copies were calculated according to the standard curve [20].

### 2.3. RNA Detection for Virus by qRT-PCR Analysis

RNA samples were extracted from mouse tissues that were flash-frozen in liquid nitrogen and stored at −80 °C or liquid nitrogen. Approximately 20 mg tissue was put into 1 mL TRIzol (Life Technologies, Carlsbad, CA, USA) and homogenized for five minutes through a homogenizer. Following the standard Thermo-Fisher protocols for TRIzol, the clear supernatant was collected, and 0.2 mL chloroform was added per 1 mL TRIzol solution. The mixture was centrifuged, and the clear supernatant retrieved. RNA was quantified using a nanodrop Spectrophotometer (Nano Drop Technologies; ND-1000, Wilmington, DE, USA) and 0.2 µg RNA was used for the reverse-transcriptase reaction using the SuperScript III RT Kit (Thermo-Fisher Scientific, Waltham, MA, USA). Following reverse transcription, 2 µL cDNA was used in the total 20 µL quantitative PCR (qPCR) SYBR Green reaction (Agilent, Santa Clara, CA, USA) according to the manufacturer’s instructions and as described in our previous studies [20,35].

### 2.4. Cytokine Profiling

A panel of host cytokines that have been reported previously to have anti-viral properties were tested in our study [25,36]. The RNA extracted from vaginal tissues harvested from either DMPA or PMSG treated mice were reverse transcribed into cDNA as described above. 2 µL cDNA of each sample was used for SYBR Green PCR analysis (Agilent, Santa Clara, CA, USA). The ratio of the fold change in transcripts of these cytokines between DMPA and PMSG treated mice was calculated (each cytokine was normalized to GAPDH).

### 2.5. Immunohistochemistry and In Situ Hybridization Analyses of Infected Tissues

After termination of the experiments, the animals were euthanized and half of each tissue of interest was frozen in liquid nitrogen and the other half fixed in 10% buffered formalin as described previously [18]. RNA ISH was performed on formalin-fixed paraffin embedded (FFPE) tissue sections using RNAscope probes (MmPV1 E4 probe) and protocols (Advanced Cell Diagnostics, ACD, Minneapolis, MN, USA). The HybEZ hybridization system of ACD was used to perform RNA-ISH. In brief, 5-μm sections were baked in the HybEZ Oven II (Advanced Cell Diagnostics; 321720, Minneapolis, MN, USA) for one hour at 60 °C and immediately deparaffinized in xylene, followed by rehydration in an ethanol series. Epitope retrieval was performed by placing the slides in RNAscope 1 × Target Retrieval Reagent (Advanced Cell Diagnostics; 322000, Minneapolis, MN, USA) at 100 °C for 15 min and then washed. Protease treatment was performed by adding RNAscope Protease Plus (Advanced Cell Diagnostics; 322331) to the section and incubating for 30 min at 40 °C in a HybEZ Oven II (Advanced Cell Diagnostics; 321720, Minneapolis, MN, USA). After probe hybridization with target probes, preamplifier and amplifier, sections were stained with Fat Brown reagent (RNAscope 2.5 HD Detection Reagents—Brown; Advanced Cell Diagnostics; 322360, Minneapolis, MN, USA). A counterstain of 50% hematoxylin and 0.02% ammonia water was used. Positive and negative probes were used in each assay to ensure proper controls. Slides from normal tissues and infected tissues were used as negative and positive controls. RNA ISH brown signals were employed under a standard bright-field microscope at 400× magnification. Representative areas of each slide were imaged at 400× magnification and positive RNA ISH signals were counted blind coded [37]. Hematoxylin and eosin (H&E) analysis, in situ hybridization (ISH) and immunohistochemistry (IHC) were conducted as described in previous studies [36]. For IHC, an in-house anti-MmuPV1 L1 monoclonal antibody (MPV.B9) and MmuPV1 anti-E4 polyclonal antibody (a generous gift from Dr. John Doorbar’s laboratory) were used on FFPE sections [38].

### 2.6. Histology and Interpretation

The samples were blindly coded and prepared as described previously. H&E staining slides were digitized and evaluated for signs of epithelial cell abnormalities by veterinary pathologists using modified Bethesda system criteria.

### 2.7. Statistical Analysis

Kruskal-Wallis tests were applied to perform similar comparisons in three or more groups, followed by Wilcoxon rank sum tests for pairwise comparisons of interest. *p* values less than 0.05 were considered statistically significant, and no adjustment was made for multiple testing because of the exploratory nature of this study. R 4.0.2 (54) was used to perform all statistical analyses. The graphs were generated in Sigma Plot 11 (Systat software, Palo Alto, CA, USA).

## 3. Results

### 3.1. DMPA Treatment Increased Viral Susceptibility at the Lower Genital and Anal Tract of Athymic Nude Mice

A previous study demonstrated that the estrous cycle played a role in genital HSV-2 infection in mice [39]. Specifically, mice at the diestrus phase showed a 100-fold increase in susceptibility to genital HSV-2 infection. In contrast, mice at the estrus stage were resistant to HSV-2 infection [39,40]. DMPA has been used to synchronize mice into a diestrus-like stage and was shown to increase HPV pseudo-virus infection at the lower genital tract [20,31]. We hypothesize that DMPA treatment increases mouse papillomavirus susceptibility at the anogenital (vaginal and anal) tracts. As a control for DMPA treatment, we synchronized the mice to an estrus-like stage using a modified protocol based on several protocols developed by other laboratories [29,41,42,43]. This protocol included two intraperitoneal doses of 0.5 µg cloprostenol, three days apart, plus a single subcutaneous dose of 3 µg progesterone coincidentally with the first injection of cloprostenol, as well as male bedding added to the female mice cages.

Three female HsD:Nu athymic nude mice (each group) were either treated with DMPA or cloprostenol/progesterone. Three days post treatment, the lower genital and anal tract were pre-wounded and subsequently infected with MmuPV1 (1 × 10^9^ viral DNA equivalent) at day four as described previously [20]. Viral DNA was monitored by collecting vaginal swabs from infected mice periodically as described previously [20]. As shown in Figure 1, significantly higher viral DNA copies were detected at the lower genital tract of DMPA treated mice when compared to the cloprostenol/progesterone (Clo-Pro) treated group starting at week 4 post infection (*p* = 0.0095, according to linear mixed model results, Figure 1A). The trend continued over time and a significant difference was found between these two groups until week 19 post infection (Figure 1A). A similar pattern was found in the anal tract of infected mice until week 15 post viral infection (*p* < 0.05, according to linear mixed model results, Figure 1B). The only difference was found at the vaginal tract but not the anal or the secondary infected sites (oral) lavages in the DMPA treated group when compared with the control group after week 19 post infection (C, *p* < 0.05, Wilcoxon rank sum tests).

### 3.2. DMPA Treatment Increased Viral Susceptibility and Persistence at the Lower Genital Tract of NU/J Heterozygous (Foxn1^nu/+^) Mice

After we demonstrated the impact of DMPA on viral susceptibility in athymic nude mice, we further investigated whether a similar effect could be observed in heterozygous NU/J (Foxn1^nu/+^) mice that have shown susceptibility to virus infection and developed advanced diseases at the lower genital tract [32]. The protocol used to synchronize the estrus-like stage for the athymic mice did not generate complete synchronization in these heterozygous NU/J (Foxn1^nu/+^) mice. We developed a second protocol using five units of pregnant mare’s serum gonadotropin (PMSG) intraperitoneal injection as well as male bedding three days before viral infection for these mice.

Ten female NU/J Foxn1^nu/+^ mice per group were either treated with DMPA or PMSG at day one. MmuPV1 (1 × 10^9^ viral DNA equivalent per infection) was delivered to each of the lower genital tract and anal tract at day 4. To mimic natural infections in humans, we infected these mice without pre-wounding, as we had demonstrated this as a valid infection method in our previous studies [20]. Consistent with what we have observed in the athymic nude mice, DMPA treated NU/J Foxn1^nu/+^ mice also showed significantly higher viral DNA copies at the lower genital tract when compared to those treated with PMSG until week nine post infection (*p* = 2.51 × 10^−5^, according to linear mixed model results, Figure 2A). The significance between DMPA and PMSG treated mice only showed in the lower genital tract, but not at the anal or the secondary oral sites, with low to undetectable viral DNA (*p* = 0.01157, *p* > 0.05, Wilcoxon rank sum test respectively, Figure 2B). These animals were sacrificed at week nine post infection and viral early RNA transcripts (E1^E4) in the infected tissues were analyzed. Consistent with data shown in lavage samples, seven out of ten DMPA treated heterozygous mice were positive for viral DNA and RNA transcripts while none of the PMSG treated animals were positive (*p* = 0.01178, Wilcoxon rank sum test). Viral DNA (Figure 2C) and RNA (Figure 2D) were significantly higher in the infected vaginal tissues of mice treated with DMPA when compared with PMSG (*p* < 0.05, Wilcoxon rank sum tests).

All the infected vaginal tissues were further analyzed for histology and viral activities. The infected athymic mice usually showed abundant viral DNA in the vaginal tract (Figure 3A, 20×). No positive viral DNA signals were found in infected heterozygous NU/J Foxn1^nu/+^ mice treated with PMSG (Figure 3B, 20×) but can be detected in the vaginal tract of mice treated with DMPA (Figure 3C, 40×). The infected heterozygous NU/J Foxn1^nu/+^ mice treated with DMPA were also positive for protein E4 (Figure 3D, 20×) and some L1 protein (Figure 3E, 20×, arrows) suggesting that productive infections occurred in these mice. Mild dysplasia was detected in the infected vaginal tissues (Figure 3F, 40×).

### 3.3. DMPA Treatment Failed to Increase Viral Susceptibility in Ovariectomized Heterozygous NU/J Mice

DMPA treatment increased viral susceptibility in both athymic nude mice and heterozygous NU/J (Foxn1^nu/+^) mice. These mice also have endogenous sex hormones that might have contributed to viral susceptibility in the DMPA treated group. To test this hypothesis, we used ovariectomized mice where endogenous sex hormones are absent. Ten ovariectomized NU/J heterozygous (Foxn1^nu/+^) female mice were divided into two groups (*n* = 5/group) and treated with either DMPA or PMSG as described in the above studies. The vaginal tract was pre wounded as described previously in order to increase the infectivity in the mice [18,19,20]. Infection status was followed by collecting lavage samples at time points post viral infection. In contrast, all DMPA and PMSG treated ovariectomized NU/J heterozygous mice were positive for viral RNA transcripts at week seven post viral infection. No significant difference was found in viral DNA (Figure 4A) copies between these two groups at week seven and eleven post infection (Figure 4A, *p* > 0.05, Wilcoxon rank sum test). No significant difference was found in viral DNA (Fig 4B) and RNA (Figure 4C) copies in the infected tissues at the time that the animals were sacrificed (*p* > 0.05, Wilcoxon rank sum test). These findings suggest that endogenous sex hormones and/or receptors played a critical role in DMPA-induced increased viral susceptibility in NU/J (Foxn1^nu/+^) mice.

### 3.4. DMPA Treatment Dysregulated Anti-Viral Activities at the Lower Genital Tissues

Previous studies suggested that DMPA might play a role in suppressing host anti-viral activities at the lower genital tract [39]. We examined the impact of DMPA on the expression of a panel of cytokines that are important innate immune modulators in the control of HPV and other pathogenic infections in humans [44,45,46]. Ten NU/J (Foxn1^nu/+^) female mice were divided into two groups (*n* = 5/group) and treated with either DMPA or PMSG. Lower genital tissues at three days post treatment were collected for RNA extraction for expression of our selected candidate genes that have been reported as playing a role in anti-viral activities (Table 1). Using GAPDH as the control, and when compared with PMSG treated animals, all selected genes were significantly downregulated in the DMPA treated animals suggesting that DMPA suppressed the host anti-viral function at the local microenvironment (Figure 5, *p* < 0.05, Wilcoxon rank sum tests).

### 3.5. Long-Term Contraceptive Treatment Did Not Promote More Advanced Diseases in Hsd:Nu Athymic Mice

Most women take contraceptives for months and years [47]. We demonstrated that one administration of DMPA increased the viral susceptibility at the lower genital tract in mice. The next question we asked was whether long-term contraceptive treatment contributes to advanced disease. In addition to DMPA, we also included another commonly used hormone, 17β-estradiol, in addition to two control groups: (1) PBS and (2) a non-hormone based contraceptive Cilostazol (CLZ, Pletal), a safe PDE3A inhibitor, that can block pregnancy without altering the estrous cycle [33,36]. Each group contained six female athymic mice that were infected with MmuPV1 (1 × 10^9^ viral DNA equivalent) at the vaginal and anal tracts respectively as described previously [18]. Four weeks after viral infection, all the mice were treated with PBS (group 1), DMPA (group 2), Cilostazol (group 3) or 17β-estradiol (group 4) for up to nine months post viral infection. Lavage samples were harvested monthly from the vaginal and anal tracts for viral DNA quantitation. As seen in Figure 6A, the treatment did not influence weight gain for any of the groups (*p* > 0.05 Wilcoxon rank sum test). Viral DNA copy numbers were similar in all groups at the vaginal tract (Figure 6B) and the anal tract (Figure 6C). We also collected oral lavages from these mice for secondary infection analyses, and no significant difference was found among the four groups (Figure 6D, *p* > 0.05, Wilcoxon rank sum test) at this latter mucosal site.

The tissues were further harvested for histological analyses. As shown in Figure 7, no significant difference was found among the four groups by H&E. One representative sample from each group at nine months after infection is presented (Figure 7). Diffuse positive mild to moderate atypia with persistent differentiation toward the lumen was found in group 1 (PBS) (Figure 7, H&E). Diffuse mild to moderate atypical hyperplasia of the caudal half of the vagina with extension into the adjacent haired skin was detected in group 2 (DMPA) and 3 (Cilostazol) (Figure 7, H&E). Adjacent to the vagina there is a tissue with florid hyperkeratosis and prominent viral CPE (interpreted as en face section of caudal vagina) (Figure 7, H&E). Rare small foci that may represent micro-invasion and desmoplasia were found in group 4 (17B-estradiol) (Figure 7, H&E). Viral DNA and RNA were detected by in situ hybridization as well as qPCR and RT-qPCR, respectively, as shown in Figure 7. No significant difference was found among the four groups (*p* > 0.05, Wilcoxon rank sum test).

Similar results were observed in the anal tissues as shown in Figure 8. No significant difference in viral DNA, RNA as well as MmuPV1E4 immunostaining was found among the groups (*p* > 0.05, Wilcoxon rank sum test).

## 4. Discussion

The contraceptive depo-medroxyprogesterone (DMPA) is an injectable form of progestin that is given every 3 months for pregnancy protection in humans. DMPA and spermicide such as N-9 have been reported to change the genital tract microenvironment and cause the vagina and cervix to be more susceptible to viral infection and persistence [48,49,50,51]. Vaginal epithelial cells of mature female mice responded to changing levels of estrogens and progesterone during a 4–5-day estrous cycle. DMPA has been used to synchronize the mice into a diestrus-like stage which showed more susceptibility to viral infections in previous studies [39,52]. Coincidently, diestrus is also highly susceptible to many viral infections in mice [40,53,54,55]. For example, a 100-fold increase in susceptibility to genital HSV-2 infection was found in mice at the diestrus phase. Mice in estrus, however, were not susceptible to HSV-2 infection [39,40]. For the mouse papillomavirus model, DMPA was also used to increase susceptibility to infection [21,22,31,56]. To understand the role of contraceptives in papillomaviral susceptibility, we tested both athymic mice (Hsd:NU) and heterozygous NU/J (Foxn1^nu/+^) mice. Pregnant mare’s serum gonadotropin (PMSG) [57,58] injection to females is widely used for treatment and increasing the number of offspring from animals, including mice, by eliminating the inhibitory mechanism of the dominant follicle or by promoting development of a subordinate follicle. For the current study, we used PMSG to induce an estrus-like stage in mice as the control for DMPA. We found that the lower genital tract became more susceptible to viral infection after DMPA treatment. Interestingly, PMSG treated athymic mice (Hsd:NU) and heterozygous NU/J (Foxn1^nu/+^) mice were less susceptible to MmuPV1 infection, suggesting that a protection against viral infection was provided by PMSG treatment.

To further understand the role of DMPA in the susceptibility of mice to MmuPV1 infection, we used ovariectomized mice so that no endogenous sex hormones were produced. These mice failed to show more susceptibility to viral infection at the lower genital tract after DMPA treatment when compared with PMSG treated mice, suggesting that endogenous sex hormones may also have played a role in the increased viral susceptibility. 17β-estradiol was reported to contribute to advanced diseases together with UV irradiation in FVB mice [21]. Estrogen and its receptor (ER) were required for cervical cancer development in K14E7 transgenic mice [59]. Two ER pathways, ERα and Erβ, are both important in tumor and cancer development [26]. For example, ERβ suppresses the expression of mini chromosome maintenance complex component 5 (MCM5), a DNA replication licensing factor that is involved in tumor cell growth [60]. Our ongoing study aims to test the hypothesis that estrogen receptors (ERα and ERβ) play a role in papillomavirus persistence and cancer development.

Dysregulation of inflammatory molecules such as cytokines was reported in HPV-associated diseases and cancers [61,62]. Our previous studies demonstrated that some innate immune pathways were involved in the susceptibility of MmuPV1 infection in the lower genital tract [63]. A recent study tested a panel of gene modified mice with deficiencies in different innate immune response pathways and found several pathways including MyD88 and Stat1 were involved [56]. We confirmed that a panel of host anti-viral immune molecules was significantly dysregulated in the lower genital tissues of mice after DMPA treatment, indicating that DMPA increased viral susceptibility by modulating these molecules. In contrast, PMSG treated mice had higher anti-viral activities in the lower vaginal tract leading to lower viral activities. Future studies will further characterize the role of DMPA and PMSG in these pathways during papillomavirus-associated infections and disease progression.

Our findings agreed with reports in the literature that progesterone hormones increase viral mRNA and significantly stimulate viral replication [39,64,65]. Studies on the relationship between contraceptive use and HPV infections have produced discordant findings in humans [27,55,66,67]. A recent study showed that contraceptives did not increase the risk of acquiring new HPV infections [27,28]. Some published studies postulate that sex hormone use could have a protective effect, depending on age: frequent acquisition of infections by different papillomavirus types may stimulate immunologic control in a young age group [67]. However, the continuous stimulatory effect of the hormones on expression of the papillomavirus genes could also lead to an increase in expression of the viral oncogenes, i.e., E6 and E7, thereby enhancing their transforming potential. Nevertheless, most studies agree that a longer duration of contraceptive use (>5 years) can be a risk for viral persistence, a key factor for cancer development [66].

We tested the long-term effect of contraceptive administration on viral persistence and advanced disease progression in one athymic mouse strain in the current study. Our study did not detect a correlation between long-term contraceptive treatment and advanced diseases in our tested mice. Several factors may be considered: (1) All mice were treated with DMPA three days before initial infection to guarantee consistent infections in all groups. Therefore, all the tested mice remained on DMPA in the early stage. Whether this treatment overshadowed different treatments among groups is unclear. (2) The mouse strain we used was outbred and had not shown advanced disease in our previous studies even though malignant conversion in skin lesions was found [17,20,36]. The nine-month treatment used in this study may be insufficient to induce malignant conversion in these athymic mice. Other mouse strains such as NU/J and FVB may be alternate choices as these strains display advanced disease including carcinoma at the lower genital tract [21,22,32]; (3) We had only six animals for each group in this pilot study. Additional studies will focus on the question as to whether long-term contraceptives promote earlier malignant conversion in the lower genital tract using NU/J athymic or heterozygous mice, as these strains have shown malignant conversion at around 9 months post infection [32].

Vaginal self-sampling has been shown to be a cost-effective method for cervical cancer screening in humans [48]. In mice, vaginal wash/swab is confirmed as a satisfactory sample collection method for both cytology and viral detection [20,49]. In the current study, we further demonstrated the dependability of lavage collection for viral detection for both the vaginal and anal tracts.

In summary, we demonstrated that DMPA increases viral susceptibility in the anogenital tract (the lower genital and anal site) of athymic nude mice and in the lower genital tract of heterozygous NU/J (Foxn1^nu/+^) mice. DMPA suppressed expression of a panel of host anti-viral cytokines in the lower genital tract of NU/J (Foxn1^nu/+^) mice. Endogenous sex hormones contributed to this increased viral susceptibility in NU/J (Foxn1^nu/+^) mice. However, long-term contraceptive treatment did not promote more advanced disease in our tested Hsd:Nu athymic mouse strain. We believe that these findings are significant as they will shed light on viral susceptibility and the role of sex hormones and contraceptives in HPV associated genital diseases and cancers.

## Figures and Tables

**Figure 1 viruses-14-00980-f001:**
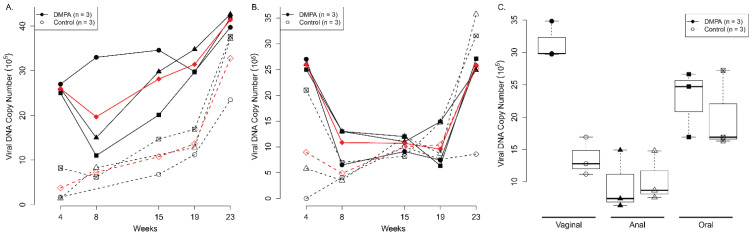
Increased viral load was detected in the lower genital tract (**A**) and anal tract (**B**) of Depo medroxyprogesterone (DMPA) treated athymic nude mice. Female athymic Hsd:Nu Mice with equivalent weight were distributed into two groups (*n* = 3) treated with DMPA (diestrus-like) or cloprostenol/progesterone (control, estrus-like stage) three days before viral infection at vaginal and anal tracts. Viral DNA was detected by collecting lavage samples from the vaginal tract, the anal tract, and the oral cavity followed by qPCR analysis at different time points post infection. Significantly higher viral DNA copies were found at the vaginal and anal site at earlier time points post infection (**A**, *p* < 0.05, Wilcoxon rank sum tests) respectively, but not the anal and the secondary infected sites (oral) lavages in DMPA treated group when compared with the control group after week 19 post infection (**C**, *p* < 0.05, Wilcoxon rank sum tests).

**Figure 2 viruses-14-00980-f002:**
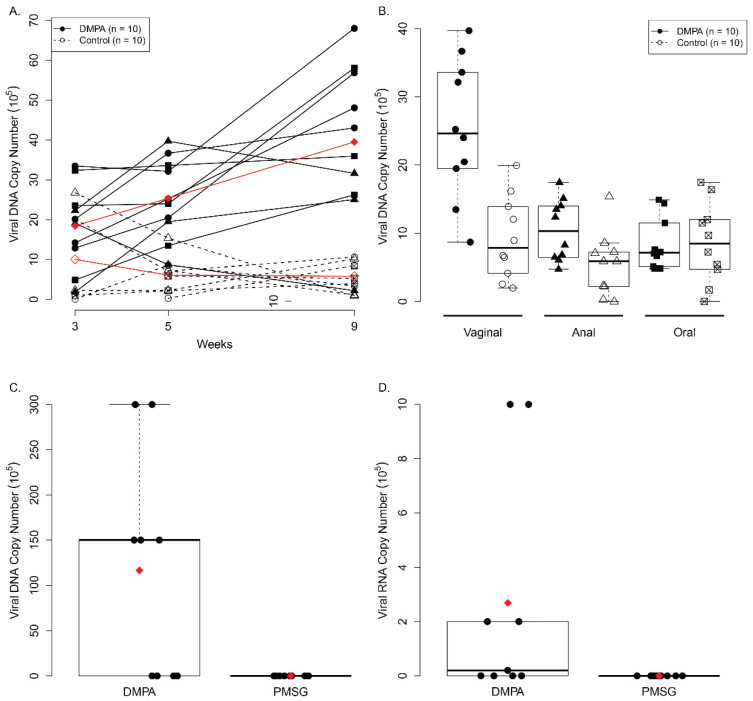
Increased viral load was detected in the lower genital tract and vaginal tissues of Depo medroxyprogesterone (DMPA) treated Nu/J heterozygous (Foxn1^nu/+^) mice. Female Nu/J heterozygous (Foxn1^nu/+^) were distributed into two groups (*n* = 10) and treated with DMPA or pregnant mare’s serum gonadotropin (PMSG) three days before viral infections at vaginal and anal tracts. Viral DNA was detected by collecting lavage samples from the vaginal tract, the anal tract, and the oral cavity followed by qPCR analysis at different time points post-infection. Significantly higher viral DNA copies were found at vaginal (**A**) but not in anal and secondary oral infected sites in the DMPA treated group when compared with the PMSG treated group up to week nine post infection (*p* < 0.05, Wilcoxon rank sum tests). Significantly higher levels of viral DNA copies were detected in DMPA treated mice while no viral DNA signal was detected in PMSG treated mice (**B**). Viral DNA (**C**) and RNA (**D**) were significantly higher in the infected vaginal tissues of mice treated with DMPA when compared with PMSG (*p* < 0.05, Wilcoxon rank sum tests).

**Figure 3 viruses-14-00980-f003:**
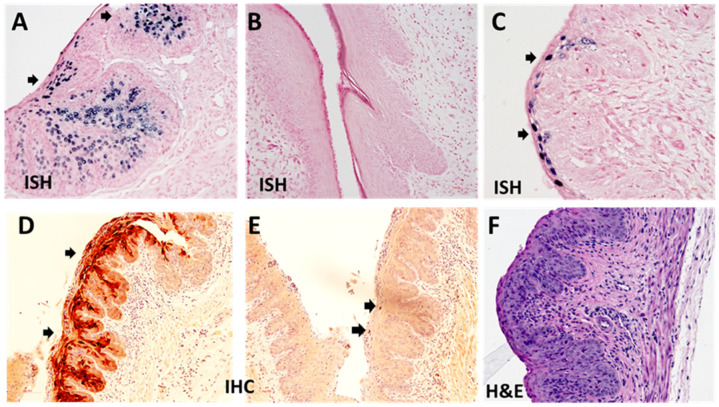
Increased viral activities were detected in the lower genital tissues of Depo medroxyprogesterone (DMPA) treated Nu/J heterozygous (Foxn1^nu/+^) mice. Vaginal tissues from female athymic HsD:Nu (**A**), Nu/J heterozygous mice (Foxn1^nu/+^) either treated with PMSG (**B**) or DMPA (**C**) and infected with MmuPV1 were further tested for viral presence by in situ hybridization (ISH, **A**–**C**)) and immunohistochemistry (IHC, **D**,**E**). Viral DNA could be readily detected in the infected vaginal tract of athymic mice (**A**, in blue as arrows pointed, 20×) but not in PMSG treated mice that were negative for viral DNA (**B**, 20×). Viral activities were also detected in the infected vaginal tissues of DMPA treated NU/J (Foxn1^nu/+^) mice (**C**, in blue as arrows pointed, 40×). The DMPA treated mice were also positive for MmuPV1 E4 (**C**, in red as arrows pointed, 20×) and capsid L1 (**C**, in blue as arrows pointed, 20×, arrows) suggesting productive infections occurred in these DMPA treated mice. Mild dysplasia was detected in the infected vaginal tissues of DMPA treated mice (**F**, 40×).

**Figure 4 viruses-14-00980-f004:**
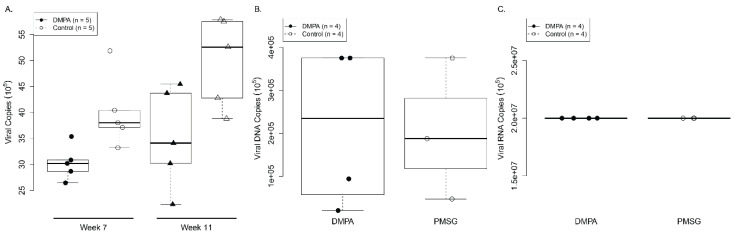
No significant difference in viral load was detected in the lower genital tract of Depo medroxyprogesterone (DMPA) treated ovariectomized Nu/J heterozygous (Foxn1^nu/+^) mice. Nu/J heterozygous (Foxn1^nu/+^) ovariectomized mice were distributed into two groups (*n* = 5 per group) and treated with DMPA or PMSG three days before viral infection at vaginal and anal sites. Viral DNA was detected by collecting vaginal lavage samples followed by qPCR analysis at different time points post infection. No significantly higher levels of viral DNA (**A**) copies were detected in DMPA treated mice when compared to those in PMSG treated mice (*p* > 0.05, Wilcoxon rank sum tests). No significantly higher levels of viral DNA (**B**) and viral RNA (**C**) copies were detected in DMPA treated vaginal tissues when compared to those in PMSG treated mice (*p* > 0.05, Wilcoxon rank sum tests).

**Figure 5 viruses-14-00980-f005:**
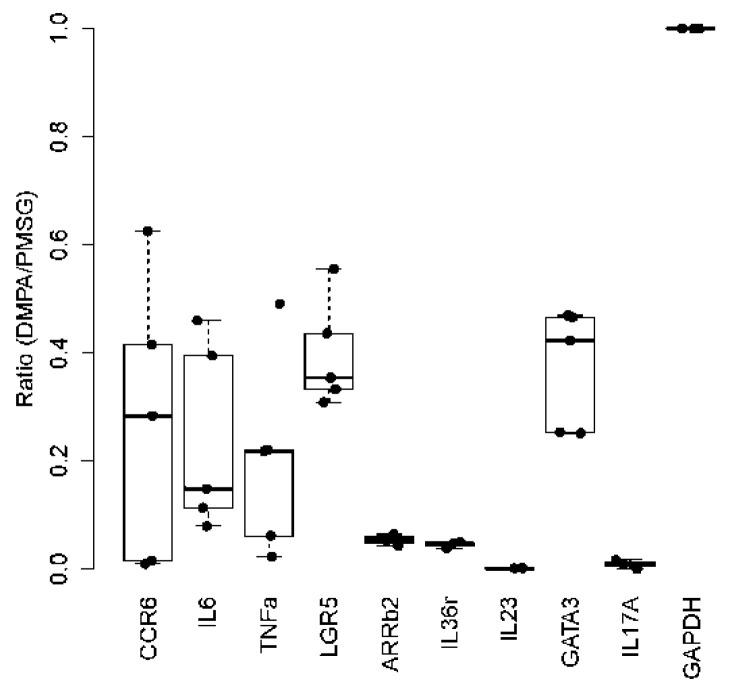
Cytokine profiling in DMPA treated vaginal tissues from Nu/J heterozygous mice (Foxn1^nu/+^). Transcripts of a panel of anti-viral cytokines including IL36γ, IL-17A, IL-6, GATA-3, TNFɑ, Arrb2 etc. were analyzed and normalized to GADPH. When compared with PMSG treated animals, significant reduction of these cytokines was found in DMPA treated vaginal tissues when compared with that in the PMSG treated group (*p* < 0.05, Wilcoxon rank sum tests).

**Figure 6 viruses-14-00980-f006:**
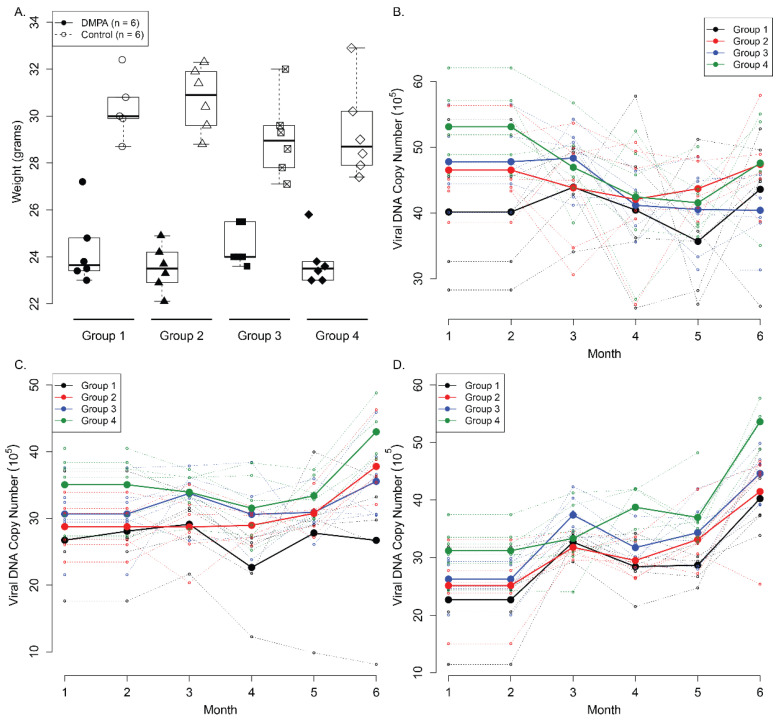
Comparable viral DNA was detected in lavages of both lower genital and anal tract in different groups. Female athymic Hsd: Nu Mice with equivalent weight were distributed into four groups (*n* = 5) treated with PBS (group 1), DMPA (group 2), Cilostazol (group 3), and17β-estradiol (group 4) at one month post viral infection and treated for up to nine months. Mice from all four groups showed even weight gain at six months post viral infection (**A**). Viral DNA was detected by collecting lavage samples following with qPCR analysis. No significant difference in viral detection was found at both vaginal (**B**) and anal (**C**) lavages among the four groups at all time points (*p* > 0.05, Wilcoxon rank sum tests). We also monitored secondary infection at the oral cavity by collecting oral lavage (**D**) and no significant difference was found (*p* > 0.05, Wilcoxon rank sum tests).

**Figure 7 viruses-14-00980-f007:**
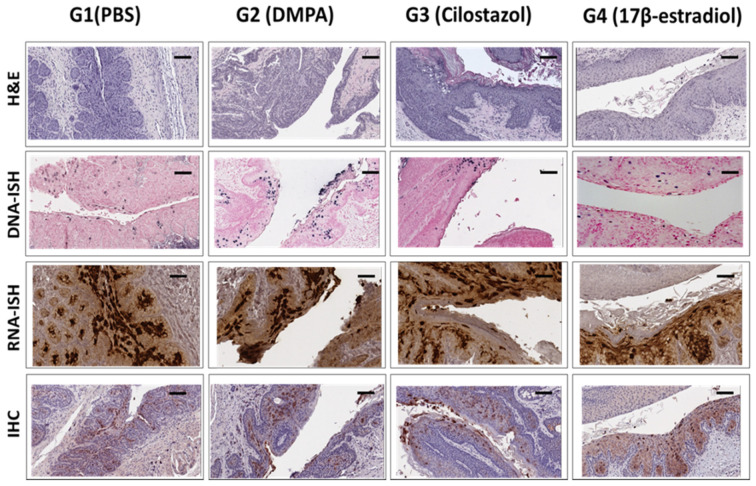
Comparable viral signals (RNA, DNA and protein) were detected in the lower genital tissues of different groups. Athymic Hsd:Nu Mice with equivalent weight were distributed into four groups (*n* = 5), treated with PBS (group 1), DMPA (group 2), Cilostazol (group 3), and17β-estradiol (group 4) at one month post viral infection and for up to nine months. Vaginal tissues were harvested for histological, virological, and immunological analyses. Representative images for each group are shown here. The top panel shows H&E staining. Diffuse positive mild to moderate atypia with persistent differentiation toward lumen was found in group 1. Diffuse mild to moderate atypical hyperplasia of the caudal half of the vagina with extension into the adjacent haired skin was found in group 2 and 3. Adjacent to the vagina there is a tissue with florid hyperkeratosis and prominent viral CPE (interpreted as an en face section of caudal vagina). Rare small foci that may represent microinvasion and desmoplasia were found in group 4. Viral DNA (the second panel), RNA (the third panel), and E4 protein (the bottom panel) were detected by in situ hybridization (ISH) and immunohistochemistry (IHC), respectively. No significant difference was found among the four groups at the termination of the experiment at nine months post treatment. Scale bar = 100 µm.

**Figure 8 viruses-14-00980-f008:**
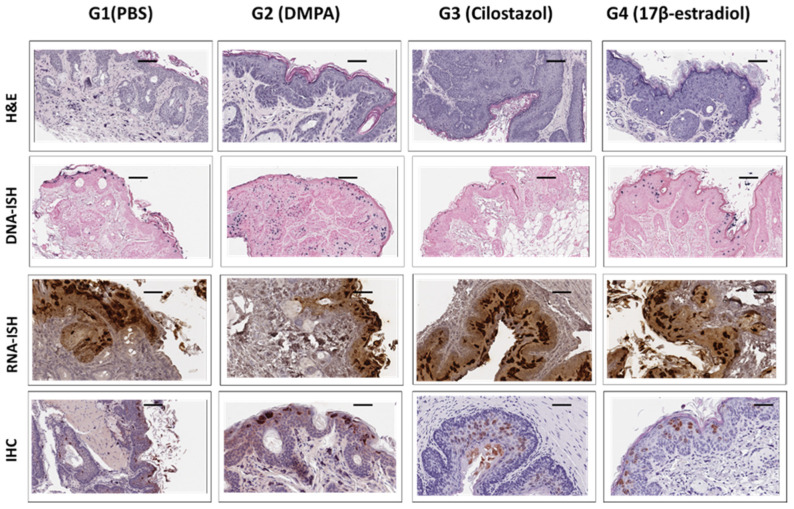
Comparable viral signals (RNA, DNA, and protein) were detected in anal tissues of different groups. Athymic Hsd:Nu Mice with equivalent weight were distributed into four groups (*n* = 5), treated with PBS (group 1), DMPA (group 2), Cilostazol (group 3), and17β-estradiol (group 4) at one month post viral infection and treated up to nine months. Anal tissues were harvested for histological, virological, and immunological analyses. Representative images for each group are shown here. The top panel shows H& E staining. Viral DNA (the second panel), RNA (the third panel), and E4 protein (the bottom panel) were detected by in situ hybridization (ISH) and immunohistochemistry (IHC), respectively. Similar histology including mild to moderate dysplasia with persistent differentiation was found in all groups. No significant difference in viral signals was found among the four groups at the termination of the experiment nine months post treatment. Scale bar = 100 µm.

**Table 1 viruses-14-00980-t001:** Primers used for cytokine profiling.

Cytokine	Sense Primer	Anti-Sense Primer
IL6	5′-AACGATGATGCACTTGCAGA-3′	5′-GGTACTCCAG AAGACCAGAG G-3′
IL36γ	5′ACCACACCCGGACAGGTGGA-3′	5′TGGGGTTGCCAGTCTTGGAGGA-3′
IL-17A	5′GCTCCAGAAGGCCCTCAGA-3′	5′AGCTTTCCCTCCGCATTGA-3′
IL-23p19	5′ACGGGGCACATTATTTTTAGTCT-3′	5′ ATGCTGGATTGCAGAGCAGTA-3′
GATA-3	5′ACCACGGGAGCCAGGTATG -3′	5′CGGAGGGTAAACGGACAGAG-3′
TNFa	5′-AG CCCCCAGTCTGTATCCTT-3′	5′-CTCC CTT TGCAGAACTCAGG-3′
ARRb2	5′-GGCAAGCGCGACTTTGTAG-3′	5′-GTGAGGGTCACGAACACTTTC-3′
LGR5	5′-TGCCCATCACACTGTCACTGT-3′	5′-CACCCTGAGCAGCATCCTG-3′
GAPDH	5′-TGGCAAAGTGGAGATTGTTGCC-3′	5′-AAGATGGTGATGGGCTTCCCG-3′

## Data Availability

Not applicable.

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
