# Peer review of "Depo Medroxyprogesterone (DMPA) Promotes Papillomavirus Infections but Does Not Accelerate Disease Progression in the Anogenital Tract of a Mouse Model"

_viruses, 2022, doi:10.3390/v14050980_

Round 1

Reviewer 1 Report

The authors investigated whether Depo medroxyprogesterone (DMPA) increases the susceptibility of MmuPV1 infection. They also investigated whether the long-term administration of DMPA induces more advanced diseases, including dysplasia in the anogenital tract, using athymic and heterozygous mice models. They confirmed the suppressed host antiviral activities by DMPA treatment as expected by other viral infection models. In addition, Ovariectomization enhanced viral infection in heterozygous mice models, suggesting the involvement of estrogen pathways in DMPA-induced susceptibility to MmuPV1 infection. However, long-term DMPA treatment did not promote more advanced diseases. They concluded that DMPA promotes initial papillomavirus infections but does not induce more advanced diseases in the infected tissues in their model systems.

  1. The reviewer thinks that the experiments and primary structure of the manuscript are well organized. However, he/she thinks that the medians should be used instead of means as representative values, and SE should be omitted in Figures 1, 2, 4, 5, and 6 because the authors used non-parametric statistical tests. They should show their data using boxplot with overlaid dot plot. Means can be used in parametric statistical analyses as representative values.

  1. Main text and Figure legends cannot be distinguished, e. g., the reviewer thinks that the lines 210-218 are legends. In addition, some explanations of the figures in the titles and the legends are missing, e. g., Figure 1C. All abbreviations also should be commented on, e.g., V, A, O in Figure 1C.

  1. Some superscript modifications are incorrect, e. g., lines 100, 230, and 337.

Author Response

Reviewer 1:

The authors investigated whether Depo medroxyprogesterone (DMPA) increases the susceptibility of MmuPV1 infection. They also investigated whether the long-term administration of DMPA induces more advanced diseases, including dysplasia in the anogenital tract, using athymic and heterozygous mice models. They confirmed the suppressed host antiviral activities by DMPA treatment as expected by other viral infection models. In addition, Ovariectomization enhanced viral infection in heterozygous mice models, suggesting the involvement of estrogen pathways in DMPA-induced susceptibility to MmuPV1 infection. However, long-term DMPA treatment did not promote more advanced diseases. They concluded that DMPA promotes initial papillomavirus infections but does not induce more advanced diseases in the infected tissues in their model systems. 

 

  1. The reviewer thinks that the experiments and primary structure of the manuscript are well organized. However, he/she thinks that the medians should be used instead of means as representative values, and SE should be omitted in Figures 1, 2, 4, 5, and 6 because the authors used non-parametric statistical tests. They should show their data using boxplot with overlaid dot plot. Means can be used in parametric statistical analyses as representative values.

We thank the reviewer for the positive comments on our project and for the suggestions on the figures. We updated these figures (F1, 2, 4, 5 and 6) using boxplot with overlaid dot plot as suggested.  

  

  1. Main text and Figure legends cannot be distinguished, e. g., the reviewer thinks that the lines 210-218 are legends. In addition, some explanations of the figures in the titles and the legends are missing, e. g., Figure 1C. All abbreviations also should be commented on, e.g., V, A, O in Figure 1C.

 We are sorry for not formatting the figure legends properly and reformatted them for easier reading. We included the description of abbreviations in the figures for Fig 1C (please see the highlighted text in Figure legend and in the result section).   

  1. Some superscript modifications are incorrect, e. g., lines 100, 230, and 337.

We corrected all superscripts for heterozygous (nu/+) as well as viral DNA (109) in the text to make them consistent and more readable.  

Reviewer 2 Report

Hu et al. examine the impact of DMPA exposure/treatment on murine papillomavirus (MmuPV1) anogenital infection and impact on tissue pathology using nude and outbred mice .

They report that DMPA promotes increased viral load in the genital tract, but not anal or oral sites vs control treated animals over the first few months. This was confirmed by copy number for the viral genome, by ISH and IHC.

These changes in viral load induced by DMPA treatment were not observed in ovariectomized mice, indicating that endogenous sex hormones may be involved in this response.  Reduced mRNA expression for a variety of cytokines was also observed in the vaginal tissues of DMPA treated mice.  No obvious changes in the pathology of infected tissues upon DMPA exposure for 9 months.

I thought this was an interesting paper and that the experiments were reasonably well done.  Clear cut differences in early infection viral loads in the genital tract are observed and reduced cytokine levels in response to DMPA may be part of the explanation. Unfortunately, this model turned out to be pretty poor for looking at the effect of DMPA on long term pathology changes as the minimal incidence of disease in these outbred mice is too low.

I was quite disappointed with the assembly and preparation of the manuscript.  The formatting is very badly messed up with respect to the figure legends.  Most of the figure legends appear as regular text in the manuscript.  There are some other minor typographical issues listed below.

Major revisions:

1) fix the figure legend formatting so that more than the first line appears to be the figure legend. The author's need to go over the legends carefully, making sure that all the details needed to interpret the figure without going back to the results are present.  For example, in the legend for F1, the first line describes panels a and b, but not c.  

2) F3, panel F needs more description.  Is this DMPA treated? Is it athymic like panel A, or heterozygous?  Should have an addition control H&E panel for equivalent, uninfected animals.

3) F5, if possible a panel for the anal or oral tissue cytokine profiles would be great for comparison.  This is not absolutely essential for publication, but would be very helpful scientifically as no changes in infection were noted in those tissues (unlike the genital data).

Minor points:

1) change 1x109 to 1x10superscript9 or 1x10E9 to be consistent with the rest of the literature in this field

2) remove authors editing comments from line 128-129

3) line 110 - ...were again anesthetized WITH ketamine....  

4) line 395- ..were distributeD into four...

5) line 433-444, this is hard to interpret as written.  please clarify

Author Response

Reviewer 2:

1. Hu et al. examine the impact of DMPA exposure/treatment on murine papillomavirus (MmuPV1) anogenital infection and impact on tissue pathology using nude and outbred mice. 

2. They report that DMPA promotes increased viral load in the genital tract, but not anal or oral sites vs control treated animals over the first few months. This was confirmed by copy number for the viral genome, by ISH and IHC. 

3. These changes in viral load induced by DMPA treatment were not observed in ovariectomized mice, indicating that endogenous sex hormones may be involved in this response.  Reduced mRNA expression for a variety of cytokines was also observed in the vaginal tissues of DMPA treated mice.  No obvious changes in the pathology of infected tissues upon DMPA exposure for 9 months. 

4. I thought this was an interesting paper and that the experiments were reasonably well done.  Clear cut differences in early infection viral loads in the genital tract are observed and reduced cytokine levels in response to DMPA may be part of the explanation. Unfortunately, this model turned out to be pretty poor for looking at the effect of DMPA on long term pathology changes as the minimal incidence of disease in these outbred mice is too low. 

We thank the reviewer for the careful reading and understanding of our experiments. As we mentioned in our discussion, we acknowledge that different mouse strains may have different pathogenesis outcomes after mouse papillomavirus infections. The long-term study in the current study was initiated before our later published paper where we observed more advanced disease in the NU/J heterozygous (nu/+) mouse strain. We therefore used the later mouse strain for viral susceptibility tests and cytokine assays. We hope to be able to test other mouse strains for a long-term study in the future when time and funds permit.  

5. I was quite disappointed with the assembly and preparation of the manuscript.  The formatting is very badly messed up with respect to the figure legends.  Most of the figure legends appear as regular text in the manuscript.  There are some other minor typographical issues listed below. 

We are sorry that we failed to assemble the manuscript better. We fixed the figure legends in the revised version and added more details on the figures to help with clarity.  

 Major revisions: 

1) fix the figure legend formatting so that more than the first line appears to be the figure legend. The author's need to go over the legends carefully, making sure that all the details needed to interpret the figure without going back to the results are present.  For example, in the legend for F1, the first line describes panels a and b, but not c.   

We thank the reviewer for the constructive comments. We have fixed the text and format of the figure legends as suggested.  

2) F3, panel F needs more description.  Is this DMPA treated? Is it athymic like panel A, or heterozygous?  Should have an addition control H&E panel for equivalent, uninfected animals. 

We thank the reviewer for pointing out the incomplete figure legends. We corrected the description in the revised figure legend (please see the highlighted text in the revision). Figure 3F is from a DMPA treated mouse. Fig 3A is from an athymic mouse as the positive control for these heterozygous tested mice.   

3) F5, if possible a panel for the anal or oral tissue cytokine profiles would be great for comparison.  This is not absolutely essential for publication, but would be very helpful scientifically as no changes in infection were noted in those tissues (unlike the genital data). 

We thank the reviewer for this excellent suggestion. Heterozygous NU/J mice did not show viral persistence (low to undetectable viral signals were found after infections) at the anal and the oral sites and therefore we did not analyze these sites for cytokines in the current study. We agree with the reviewer that this additional comparative data may provide some insights in how DMPA modulates local immune responses in different sites. We are currently working on another manuscript focusing on oral infections and will consider analyzing cytokine profiles.  

Minor points: 

1) change 1x109 to 1x10superscript9 or 1x10E9 to be consistent with the rest of the literature in this field 

We changed 1x109 to 1x109 throughout the manuscript.  

2) remove authors editing comments from line 128-129 

We removed the comments. 

3) line 110 - ...were again anesthetized WITH ketamine....   

We added “with” after anesthetized 

4) line 395- ..were distributeD into four... 

We added “d” after “distribute” 

5) line 433-444, this is hard to interpret as written.  please clarify 

We rewrote the paragraph to make it more reader friendly. These are pathological terms that our pathologist provided for us.